# Assessing clinical quality performance and staffing capacity differences between urban and rural Health Resources and Services Administration-funded health centers in the United States: A cross sectional study

Nadereh Pourat[1,2]*, Xiao Chen[1], Connie Lu[1], Weihao Zhou[1], Hank Hoang[3], Alek Sripipatana[3]

1 Center for Health Policy Research, University of California, Los Angeles (UCLA), Los Angeles, California, United States of America, 2 Department of Health Policy and Management, Fielding School of Public Health, UCLA, Los Angeles, California, United States of America, 3 Bureau of Primary Health Care, Health Resources and Services Administration, U.S. Department of Health and Human Services, Rockville, Maryland, United States of America

* pourat@ucla.edu

## Abstract

### Background

In the United States, there are nearly 1,400 Health Resources and Services Administration-funded health centers (HCs) serving low-income and underserved populations and more than 600 of these HCs are located in rural areas. Disparities in quality of medical care in urban vs. rural areas exist but data on such differences between urban and rural HCs is limited in the literature. We examined whether urban and rural HCs differed in their performance on clinical quality measures before and after controlling for patient, organizational, and contextual characteristics.

### Methods and findings

We used the 2017 Uniform Data System to examine performance on clinical quality measures between urban and rural HCs (n = 1,373). We used generalized linear regression models with the logit link function and binomial distribution, controlling for confounding factors. After adjusting for potential confounders, we found on par performance between urban and rural HCs in all but one clinical quality measure. Rural HCs had lower rates of linking patients newly diagnosed with HIV to care (74% [95% CI: 69%, 80%] vs. 83% [95% CI: 80%, 86%]). We identified control variables that systematically accounted for eliminating urban vs. rural differences in performance on clinical quality measures. We also found that both urban and rural HCs had some clinical quality performance measures that were lower than available national benchmarks. Main limitations included potential discrepancy of urban or rural designation across all HC sites within a HC organization.

**Data Availability Statement:** All relevant data are within the manuscript and its Support information files.

**Funding:** This article was funded by the U.S. Department of Health and Human Services, Health Resources and Services Administration (HRSA, https://www.hrsa.gov/) under HRSA Contract number HHSH250201300023I (NP). The views expressed in this article are solely the opinions of the authors and do not necessarily reflect the official policies of the U.S. Department of Health and Human Services or HRSA, nor does mention of the department or agency names imply endorsement by the U.S. Government.

**Competing interests:** The authors have read the journal's policy, and the authors of this manuscript have the following competing interests to declare: HH and AS are employees of the U.S. Government, U.S. Department of Health and Human Services, which funded this study. This does not alter our adherence to PLOS ONE policies on sharing data and materials.

**Abbreviations:** AHRF, Area Health Resource File; BMI, body mass index; CI, confidence interval; FTE, full-time equivalent; HC, health center; HIV, human immunodeficiency virus; HRSA, Health Resources and Services Administration; PCP, primary care provider; SD, standard deviation; UDS, Uniform Data System.

## Conclusions

Findings highlight HCs' contributions in addressing rural disparities in quality of care and identify opportunities for improvement. Performance in both rural and urban HCs may be improved by supporting programs that increase the availability of providers, training, and provision of technical resources.

## Introduction

An estimated 72 percent of the land area in the United States is considered rural and 14 percent of the population or 46.1 million people live in rural areas [1]. Evidence indicates significant and persistent rural disparities in quality of care. Rural areas (micropolitan and noncore) were found to have worse quality of care based on performance on 250 quality indicators examined compared to large suburban areas [2]. In addition, these disparities have remained the same for over one third or worsened for about one in ten of these indicators from 2000 to 2015 [2]. The indicators with worse quality of care constituted about a third of the 26 effective treatment indicators, a third to half of the care coordination indicators, and a quarter to one third of the 18 access to care indicators in rural areas [2]. These disparities co-occur with variations in sociodemographics and disparities in health status [3–6]. Nationally, rural populations are more socioeconomically vulnerable than urban areas. This includes more often being without high school education (16% vs. 13%), low-income (38% vs. 33% under federal poverty guidelines), with Medicaid coverage (24% vs. 22%), and in poorer health (13% vs. 9%). It also includes being less often employed (56% vs. 58%), and female (48% vs 50%) [7–9]. In addition, 85% of rural counties in the United States were persistently considered primary care Health Professional Shortage Areas at least once from 1996 to 2005, which impacts access to care and might impact clinical quality measure performance [10, 11].

Health centers funded by the Health Resources and Services Administration (HRSA) (referred to as HCs hereon) have a significant presence in rural areas and may be the only providers in some rural locations [3]. In some areas, more than 20% of the low-income population are served by HCs [12]. HCs are also mandated as part of their mission to provide care to those who are geographically isolated, including those in rural settings. In 2017, about 44% of the 1,373 HCs served rural populations, operated more than 4,400 sites, and collectively provided care through 35 million visits to more than 27 million total patients [13]. Earlier studies of HCs indicated that rural HC patients were more often older, female, White, poor, uninsured, obese, in poor health, and with activity limitations compared to the general rural population [3]. In addition, rural HCs had lower staffing supply for primary, mental health, and dental care providers than urban settings [3, 4, 6, 14, 15]. Strategic quality improvement initiatives over the years have improved provider recruitment and retention in rural areas and may have reduced disparities in access to care and quality of care [16, 17]. In fact, past research assessing performance on clinical quality at HCs has shown that rural HCs performed well in prenatal care outcomes, cervical cancer screening, and childhood immunization rates [3, 18, 19].

Existing data on urban/rural disparities in quality of care among HCs is limited, and a gradual decline in the size of rural populations may have influenced previous understanding of this issue [1]. Recent literature has found that despite adjusting for individual and community-level factors, several differences in quality of care measures were observed between urban and rural general populations [20, 21]. These findings suggest other unknown mechanisms may be affecting quality of care among urban and rural areas. A comprehensive assessment of the clinical

quality of HCs in rural communities is necessary to identify areas in need of improvement and reduce missed opportunities in addressing rural disparities in quality of care. To address this gap, we examined differences in performance on clinical quality measures between rural and urban HCs, controlling for differences in characteristics of these organizations as well as other contextual factors that might impact performance on these measures. We hypothesized that rural HCs performed as well as urban HCs because of the emphasis HRSA places on improving access to quality health care services for all HCs. HRSA has supported quality improvement by requiring reporting on performance, incentivizing improvement and meeting national benchmarks of performance and providing financial and technical support to improve performance [22–24]. Our study aimed to highlight disparities in performance between rural and urban HCs and to identify factors that may mediate urban/rural performance variations.

## Methods

### Data and sample

For this cross-sectional study, we used data from the 2017 Uniform Data System (UDS) reported by the entire HRSA-funded HC patient population on organizational characteristics and clinical quality measures. UDS is an administrative data source maintained by HRSA to monitor the Health Center Program and provide information to stakeholders. HRSA-funded HCs and other entities receiving federal funding authorized under Section 330 of the Public Health Service Act are required to report UDS data. UDS captures aggregate information at the organization level rather than individual delivery sites that operate within the organization [22].

We merged the UDS data with the latest available relevant data from the 2016 Area Health Resource File (AHRF). AHRF is a publicly available dataset maintained by HRSA. It compiles information from over 50 sources to provide county-level data on population characteristics, health workforce availability, health care utilization, and health facilities [25]. We merged these data using the Federal Information Process Standards (FIPS) code associated with the address of the HC organization. If HCs were present in multiple counties, we merged the data for the county where the largest share of HC patients lived. We included all HCs that reported serving patients in 2017 for a total analytic sample of 1,373.

### Dependent variables

We studied 15 clinical quality measures that HCs are required to report. The majority of these measures were standard quality metrics and concordant with Centers for Medicare and Medicaid Services guidelines and electronically specified for automated reporting by HCs (electronic clinical quality measures) [26]. These measures had national benchmarks included in the 2017 Healthcare Effectiveness Data and Information Set (HEDIS) for Medicaid Managed Care patients [27, 28]. All 15 measures were included in our analyses, with seven measures examining prevention-related performance, five measures associated with care management, and three measures assessing clinical outcomes [29, 30]. The seven prevention measures included (1) up-to-date childhood immunization completion, (2) receipt of recommended cervical cancer screening, (3) receipt of colorectal cancer screening, (4) tobacco use and cessation counseling and intervention, (5) depression screening and receipt of a follow-up plan, (6) weight assessment and counseling for nutrition and physical activity for children and adolescents, and (7) body mass index (BMI) screening and follow-up plan for adults. The five measures examining aspects of care management included: (1) patients with asthma receiving appropriate medications, (2) patients with coronary artery disease that were prescribed lipid-lowering therapy, (3) patients with ischemic vascular disease who used aspirin or another antithrombotic drug, (4) patients seen for follow-up care within 90 days of initial HIV diagnosis, and (5) pregnant

women who received early prenatal care. Three measures examined outcomes of care, including (1) patients with diabetes whose hemoglobin A1c level was greater than 9% (poorly controlled), (2) patients with diagnosed hypertension whose blood pressure was below 140/90 (controlled), and (3) patients born whose birthweight was below normal (2,500 grams). The full measure definitions are described in S1 Table.

We created a dichotomous indicator variable that measured the proportion of urban and rural HCs that met or exceeded (vs. did not) the 2017 HEDIS national benchmarks for each clinical quality measure.

### Independent variables

The primary variable of interest was urban/rural status. HCs self-reported this status for the organization and used the same designation for all delivery sites. In some instances, HCs had several service delivery sites across both urban and rural areas, which resulted in some misclassification. We controlled for several HC organizational characteristics, patient characteristics, and contextual variables to account for any potential confounding among HC and local area factors. The HC organizational characteristics controlled for included the organizational size indicated by the number of sites and number of patients seen in 2017. We further controlled for patient demographic and health characteristics including percent of patients ages 0–17 and ages 65 or older, patients who were racial/ethnic minorities, patients who communicated with the provider in a language other than English, patients with heart related disease, patients with diabetes or endocrine disease, patients with respiratory disease, patients with HIV, and prenatal care patients who delivered during the year. These variables controlled for HC case mix and challenges to care outcomes.

We next controlled for primary care and other care capacity and service availability using several indicators. These included the ratio of HC patients per each full-time equivalent (FTE) primary care provider (PCPs include physicians, nurse practitioners, and physician assistants) and ratio of FTE nurses per PCP, the ratio of mental health providers (psychiatrists, psychologists, licensed clinical social workers, other licensed mental health providers) per 5,000 patients, dental providers (dentists and hygienists) per 2,500 patients, enabling service staff (case managers, transportation, and translation staff) per 5,000 patients, and an index of number of services available in addition to medical care. We also included financial resource indicators, including per capita total revenues to measure success in generating revenues and proportion of grant revenues from the Section 330 grants to measure success in fundraising. The contextual control variables, extracted from AHRF, included the ratio of PCP per 5,000 individuals in the county, the proportion of individuals below the federal poverty guideline, and the proportion of minorities in the county.

### Statistical analysis

We compared the independent and control variables by urban/rural status using t-tests. We then developed fractional outcome regression models using the fracreg command and logit distribution [31, 32]. These models were used to compare clinical quality measures after adjusting for confounding impact of HC patient and organizational characteristics and county-level contextual factors. We further compared the proportion of urban vs. rural HCs that met or exceeded 2017 HEDIS benchmarks using logistic regression models and adjusting for control variables. We included only complete data for all analyses presented in this paper. All HCs were treated with equal analytical weight. All analyses were conducted using Stata v. 15, and we used the Margins command to report predicted probabilities for ease of interpretation. We discussed all statistically significant results with probability values of 0.05 or smaller.

### Ethics statement

Secondary data on HCs were de-identified, and as such, the study was granted written exemption from review by the University of California Los Angeles Institutional Review Board (study number 16–001528).

## Results

Table 1 indicated about 44% of HCs were rural. On average, rural HCs were smaller than non-rural HCs as indicated by fewer sites (7.4 [SD 8.1] vs. 8.6 [SD 9.6]) and patients (14,673 [SD 18,068] vs. 23,861[SD 26,623]). Rural HCs had more older patients (13.2% [SD 6.6%]vs 7.2% [SD 4.2%]), fewer racial/ethnic minorities (38.2% [SD 31.1%] vs 70.7% [SD 23.5%]), and less Medicaid patients (35.1% [SD 17.1%] vs. 50.7% [SD 18.6%]) than urban HCs. Rural HCs also had more patients with heart and respiratory diseases but fewer patients with HIV or prenatal patients who delivered than urban HCs. Rural HCs also differed from urban HCs in capacity of with a higher ratio of dental providers per 2,500 patients (0.8 [SD 0.8] vs 0.7 [SD 0.7]), and nurses to PCPs (0.9 [SD 0.6] vs. 0.7 [SD 0.5]), but lower PCP panel size (1,092 [SD 514] vs. 1,168 [SD 501]) and ratio of mental health providers to 5,000 patients (1.6 [SD 2.1] vs 2.2 [SD 3.9]). Section 330 grants represented a higher percentage of total revenue in rural HCs than urban HCs (34.0% [SD 17.9%] vs. 25.9% [SD 17.9%]). Rural HCs also had lower PCP capacity in the county overall and less racial/ethnic diversity but more patients living in poverty than urban HCs.

Unadjusted clinical quality measures showed multiple differences between rural and urban HCs including lower rates of up-to-date child immunizations (30% [SD 23%] vs 38% [SD 23%]), recommended cervical cancer screening (47% [SD 17%] vs. 53% [SD 18%]), and weight assessment and counseling for nutrition and physical activity for children and adolescents (55% [SD 26%] vs. 62% [SD 26%], Fig 1). After adjusting for patient, organizational and county-level characteristics, there were no differences between rural and urban HCs among prevention measures. Among unadjusted care management measures, rural HCs had lower rates of appropriate pharmacological therapy for patients with persistent asthma (83% [SD 17%] vs. 86% [SD 13%]), lipid lowering therapy for patients with coronary artery disease (78% [SD 15%] vs. 80% [SD 13%]), use of antithrombotic drugs for patients with ischemic vascular disease (76% [SD 16%] vs. 78% [SD 14%]), and linkage to care for newly diagnosed HIV patients (71% [SD 40%] vs. 84% [SD 27%]) but higher rates of early prenatal care for pregnant patients (81% [SD 16%] vs. 74% [SD 15%]) compared to urban HCs. After adjustment, the differences in clinical quality measures remained statistically significant only in care management of newly diagnosed HIV patients. Rural HCs had a predicted probability of 75% [95% CI: 69%, 80%] of newly diagnosed HIV patients being linked to care in 90 days compared to 83% [95% CI: 80%, 86%] in urban HCs. Among unadjusted outcome quality measures, rural HCs had different performance rates, with 32% (SD 12%) of rural HCs reporting patients with diabetes had uncontrolled hemoglobin A1c levels (vs. 35% [SD 12%] of urban HCs) and 63% reporting patients with hypertension had their blood pressure controlled (vs. 61% of urban HCs). All the unadjusted and adjusted clinical quality measure outcomes are presented in S2 Table.

Differences in all clinical quality measures, with the exception of 90-day follow-up care for newly diagnosed HIV patients, were explained by underlying differences in patient demographics and health status, organizational characteristics, and contextual factors to varying degrees and depending on the performance measure. For example, the difference between urban and rural HC performance on childhood immunization completion was explained by the higher number of non-English speaking patients, higher rate of children at the HC, and

**Table 1. Contextual characteristics.**

| | Total | Urban | Rural | P-value (Urban vs. Rural) |
|---|---|---|---|---|
| **Sample Size** n (%) | 1,373 | 765 (56%) | 608 (44%) | |
| | Mean (SD) | Mean (SD) | Mean (SD) | |
| *Organization Size* | | | | |
| Average number of sites | 8.1 (9.0) | 8.6 (9.6) | 7.4 (8.1) | 0.010 |
| Average number of patients seen during the year | 19,792 (23,663) | 23,861 (26,623) | 14,673 (18,068) | 0.000 |
| *Patient Characteristics/Complexity* | | | | |
| Percent of patients 65 years and older | 9.8% (6.2%) | 7.2% (4.2%) | 13.2% (6.6%) | 0.000 |
| Percent of patients between 0 to 17 years old | 26.4% (12.9%) | 26.9% (14.3%) | 25.8% (10.7%) | 0.130 |
| Percent of patients that were racial/ethnic minorities | 56.3% (31.6%) | 70.7% (23.5%) | 38.2% (31.1%) | 0.000 |
| Percent of patients that spoke with primary care provider (PCP) in a language other than English | 19.1% (22.8%) | 24.3% (22.0%) | 12.5% (22.2%) | 0.000 |
| Percent of Medicaid patients | 43.8% (19.5%) | 50.7% (18.6%) | 35.1% (17.1%) | 0.000 |
| Percent of patients with heart related disease | 3.1% (2.1%) | 2.5% (1.6%) | 3.9% (2.4%) | 0.000 |
| Percent of patients with diabetes or endocrine diseases | 9.5% (4.2%) | 9.3% (4.1%) | 9.6% (4.2%) | 0.280 |
| Percent of patients with respiratory diseases | 3.3% (2.4%) | 2.5% (1.8%) | 4.3% (2.7%) | 0.000 |
| Percent of patients with HIV | 0.8% (3.2%) | 1.3% (4.1%) | 0.2% (1.3%) | 0.000 |
| Percent of prenatal care patients who delivered during the year | 0.8% (0.9%) | 1.0% (1.0%) | 0.6% (0.8%) | 0.000 |
| *PCP Staffing and Capacity* | | | | |
| PCP Panel Size (Patients per PCP) | 1134.8 (508.1) | 1168.6 (501.0) | 1092.0 (514.1) | 0.006 |
| Ratio of nurses to PCP | 0.8 (0.6) | 0.7 (0.5) | 0.9 (0.6) | 0.000 |
| *Additional Staffing and Capacity* | | | | |
| Ratio of mental health provider per 5,000 patients | 1.9 (3.2) | 2.2 (3.9) | 1.6 (2.1) | 0.000 |
| Ratio of dental provider per 2,500 patients | 0.8 (0.7) | 0.7 (0.7) | 0.8 (0.8) | 0.004 |
| Ratio of enabling service staff per 5,000 patients | 5.3 (7.4) | 6.0 (8.5) | 4.5 (5.6) | 0.000 |
| Average number of services provided in addition to medical care | 3.5 (1.6) | 3.6 (1.6) | 3.3 (1.6) | 0.000 |
| *Financial Resources* | | | | |
| Per capita total revenues | $1,084 ($928) | $1,112 ($890) | $1,048 ($973) | 0.207 |
| Percent of total revenues that are from 330 grants | 29.5% (18.7%) | 25.9% (17.9%) | 34.0% (18.6%) | 0.000 |
| *Contextual characteristics* | | | | |
| Ratio of PCP per 5,000 population in county | 3.8 (1.7) | 4.3 (1.4) | 3.2 (1.8) | 0.000 |
| Percent below federal poverty guideline in county | 16.1% (5.6%) | 15.3% (4.8%) | 17.2% (6.3%) | 0.000 |
| Percent of minority in county | 38.8% (23.7%) | 46.8% (20.8%) | 28.5% (23.2%) | 0.000 |

Standard deviation in parentheses. Analyses involved comparing independent and control variables by urban and rural health center status using t-tests.

PCP, Primary care provider; SD, standard deviation.

lower rate of patients with respiratory diseases (S3 Table). We also compared the predicted probabilities of each measure for urban and rural HCs with HEDIS Medicaid Managed Care national benchmarks and found that on average rural HCs met or exceeded the benchmarks for the preventive measures of tobacco use and cessation counseling and intervention, care management measures of patients with asthmas receiving appropriate medications and lipid lowering therapy for patients with coronary artery disease, and outcome measures of patients with diabetes with hemoglobin A1c greater than 9% and patients with hypertension with blood pressure below 140/90mmHg (Table 2).

We further examined the predicted probability of proportion of urban and rural HCs that met or exceeded these HEDIS benchmarks, when benchmarks were available. These data

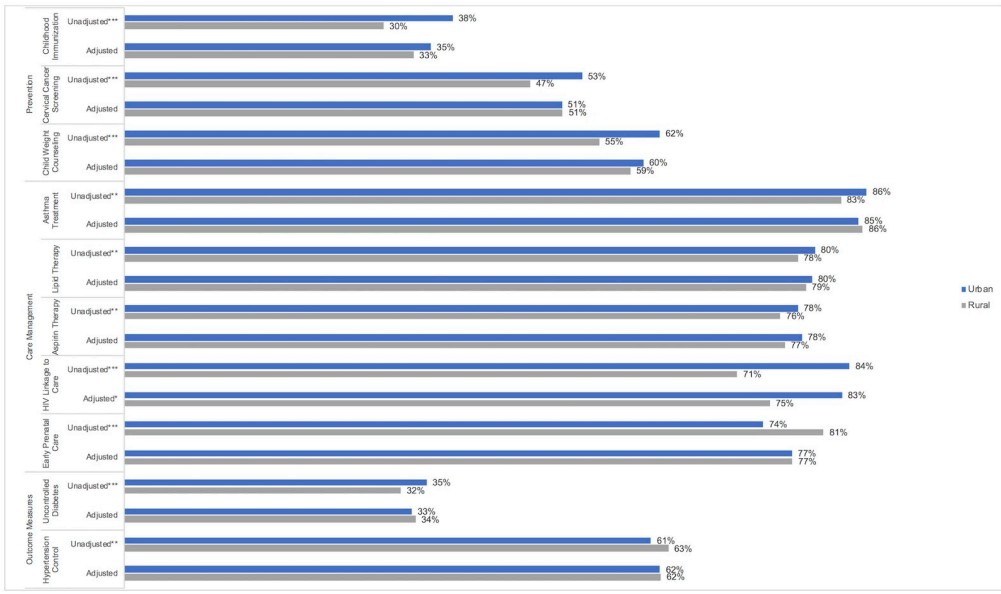

**Fig 1. Unadjusted and adjusted predicted probabilities for health center quality indicators by urban and rural status.** Unadjusted analyses involved comparing urban and rural health center status using t-tests. Adjusted analyses were conducted using fractional outcome regression models using the logit distribution. Statistically significant at *p<0.05; **p<0.01; ***p<0.001 comparing urban vs. rural.

showed that predicted probabilities of proportion of urban and rural HCs that met or exceeded the preventive, care management, and outcome measures were statistically similar, despite apparent differences. For example, 48% of urban and 41% of rural HCs met or exceeded the national benchmark of 35% for up-to-date childhood immunization completion rate and this apparent difference was not statistically significant. The full logistic regression models are displayed in S6–S8 Tables.

The control variables that explained differences in urban/rural HC performance are displayed in Table 3. These results showed that performance differences in preventive measures were explained by proportion of non-English speaking patients, percentage of patients with diabetes, percentage of patients 0–17 years of age, and ratio of PCPs per 5,000 persons in the county. In contrast, contextual factors such as percentage of poor or minority patients in the county did not predict differences in preventive measures. Urban/rural differences in care management measures were most frequently explained by percentage of non-English speaking patients, but other characteristics did not play a major or any role. Finally, urban/rural differences in outcome measures were most frequently explained by percentage of patients who were racial/ethnic minorities and percentage who were non-English speaking patients. The remaining variables played a role less frequently or did not have a role. The full regression models with the coefficients for each control variable are noted in S3–S5 Tables.

## Discussion

Our findings show that rural HCs had significant differences in patient, organizational, and contextual characteristics compared with urban HCs. Rural HCs also had lower performance on preventive and care management measures but better performance on outcome measures than urban HCs, though these differences were small. We found that nearly all urban-rural clinical quality measure differences could be attributed to patient, organizational, and

**Table 2. Predicted probabilities of proportion of urban and rural health centers that met or exceed quality benchmarks.**

| Measure Definition | 2017 HEDIS Medicaid Managed Care Benchmarks[1] | Predicted probabilities [2] | | | |
| --- | --- | --- | --- | --- | --- |
| | | Urban | | Rural | |
| Sample Size *n (%)* | | 765 (56%) | | 608 (44%) | |
| | | *Predicted Probability* | *95% CI* | *Predicted Probability* | *95% CI* |
| **Prevention** | | | | | |
| Childhood Immunization | 35% | 48% | [44%,53%] | 41% | [36%,46%] |
| Cervical Cancer Screening | 59% | 32% | [28%,36%] | 33% | [28%,37%] |
| Tobacco Use Counseling | 77% | 84% | [81%,87%] | 80% | [76%,84%] |
| Child Weight Counseling | 73% | 38% | [34%,42%] | 35% | [29%,40%] |
| Adult Body Mass Index (BMI) Screening | 85% | 19% | [15%,23%] | 17% | [13%,21%] |
| **Care Management** | | | | | |
| Asthma Treatment | 61% | 94% | [92%,97%] | 93% | [91%,96%] |
| Lipid Therapy | 76% | 69% | [65%,73%] | 67% | [63%,72%] |
| Early Prenatal Care | 81% | 44% | [40%,48%] | 45% | [40%,50%] |
| **Outcomes** | | | | | |
| Uncontrolled Diabetes | 41% | 79% | [76%,82%] | 80% | [76%,84%] |
| Hypertension Control | 57% | 71% | [67%,74%] | 73% | [68%,77%] |

Notes:

[1] Benchmarks are based on national benchmarks in the 2017 Healthcare Effectiveness Data and Information Set for Medicaid Managed Care patients.

[2] Analyses were conducted using logistic regression models and creating an indicator outcome variable of whether the health center met or exceeded the 2017 HEDIS Medicaid Managed Care Benchmarks.

[3] Several clinical quality measures did not have an associated 2017 HEDIS Medicaid Managed Care Benchmark. These included colorectal cancer screening, depression screening and follow-up, aspirin therapy, HIV linkage to care, and low birth weight.

Standard deviation in parentheses.

BMI, body mass index; CAD, coronary artery disease; IVD, ischemic vascular disease; HIV, human immunodeficiency virus; HbA1c, Hemoglobin A1c; SD, standard deviation; HC, health center; HEDIS, Healthcare Effectiveness Data and Information Set; CI, confidence interval.

contextual differences, with varying characteristics as the explanatory factors for performance differences on specific clinical quality measures.

For example, the higher percentage of patients who preferred care provided in a language other than English was associated with better performance measures across the board, with the exception of routine depression screening. This finding may indicate the value of delivering linguistically and culturally competent care. Enabling services staff in HCs provide translation services that can improve care outcomes because patients may better understand provider instructions [33, 34].

Similarly, we found a positive association for six of the seven measures between higher percentage of patients with diabetes and better preventive performance [35]. HCs focusing on improving diabetes outcomes may target diabetes patients for additional opportunities for comprehensive preventive care services (i.e., earlier pneumonia vaccines, weight screening, diet counseling) and patients with diabetes may visit HCs more frequently and therefore have more opportunities to receive preventive care [34, 36, 37]. Similarly, the positive association of higher percentages of younger patients with better child and adult preventive performance measures (five of seven) is likely because HCs with younger patients focused on provision of such services to children and their parents [38, 39]. In some instances, such a positive relationship between more diabetes patients with lower rates of poorly controlled diabetes and better

**Table 3. Select significant health center characteristics and organizational factors associated with clinical performance measures.**

| | Prevention | | | | | | | | Care Management | | | | Outcomes | | |
|---|---|---|---|---|---|---|---|---|---|---|---|---|---|---|---|
| | Up-to-Date Childhood Immunization Completion | Receipt of Recommended Cervical Cancer Screening | Receipt of Colorectal Cancer Screening | Tobacco Use and Cessation Counseling and Intervention | Depression Screening and Receipt of a Follow-up Plan | Weight Assessment and Counseling for Nutrition and Physical Activity for Children and Adolescents | Body Mass Index (BMI) Screening and Follow-up Plan for adults | Patients with Asthma Receiving Appropriate Medications | Patients with Coronary Artery Diseases That Were Prescribed Lipid-Lowering Therapy | Patients with Ischemic Vascular Disease Who Used Aspirin or Another Antithrombotic Drug | Patient Seen for Follow-up Care within 90 Days of initial HIV Diagnosis | Pregnant Women Who Received Early Prenatal Care | Patients with Diabetes with Hemoglobin A1c Greater Than 9% | Patients with Hypertension with Blood Pressure below 140/90 | Patients Born Whose Birthweight Was Below Normal |
| **Urban (vs. rural)** | | | | | | | | | | | ← | | | | |
| ***Organization Size*** | | | | | | | | | | | | | | | |
| Average number of sites | | | | | | ← | | | | | | → | | | → |
| Average number of patients seen during the year | | ← | ← | | ← | | | | | | | ← | ← | | |
| ***Patient Characteristics*** | | | | | | | | | | | | | | | |
| Percent of patients that were racial/ethnic minorities | ← | | ← | | | | | | | | | ← | ← | | |
| Percent of patients that spoke with primary care provider (PCP) in a language other than English | | ← | ← | ← | | ← | ← | | | | | | | | → |
| Percent of patients 65 years and older | | | → | → | | | | | | → | | → | → | → | ← |
| Percent of patients between 0–17 years | ← | ← | ← | ← | ← | ← | | ← | ← | ← | ← | ← | ← | ← | → |
| Percent of patients with heart related disease | | | | | | | | | ← | | | | | ← | |
| Percent of patients with diabetes or endocrine diseases | | | ← | | | | | | | | | | | | |
| Percent of patients with respiratory diseases | | ← | ← | ← | ← | ← | | | | | | | ← | ← | |
| Percent of patients with HIV | → | → | → | | | ← | ← | | | | | | → | | |

*(Continued)*

**Table 3.** (Continued)

| | Prevention | | | | | | | Care Management | | | | | Outcomes | | |
|---|---|---|---|---|---|---|---|---|---|---|---|---|---|---|---|
| | Up-to-Date Childhood Immunization Completion | Receipt of Recommended Cervical Cancer Screening | Receipt of Colorectal Cancer Screening | Tobacco Use and Cessation Counseling and Intervention | Depression Screening and Receipt of a Follow-up Plan | Weight Assessment and Counseling for Nutrition and Physical Activity for Children and Adolescents | Body Mass Index (BMI) Screening and Follow-up Plan for adults | Patients with Asthma Receiving Appropriate Medications | Patients with Coronary Artery Diseases That Were Prescribed Lipid-Lowering Therapy | Patients with Ischemic Vascular Disease Who Used Aspirin or Another Antithrombotic Drug | Patient Seen for Follow-up Care within 90 Days of initial HIV Diagnosis | Pregnant Women Who Received Early Prenatal Care | Patients with Diabetes with Hemoglobin A1c Greater Than 9% | Patients with Hypertension with Blood Pressure below 140/90 | Patients Born Whose Birthweight Was Below Normal |
| Percent of prenatal care patients who delivered during the year | | | | | | | ↑ | | ↑ | | ↑ | | ↑ | | |
| Percent of Medicaid Patients | | ↑ | ↑ | | | ↑ | | | ↑ | | | ↓ | | | ↓ |
| **_PCP Staffing and Capacity_** | | | | | | | | | | | | | | | |
| PCP Panel Size (Patients Per Provider) | | | | | ↑ | ↑ | ↓ | | ↑ | | | | | | |
| Ratio of nurses to PCP | | | | | | | | | | ↑ | | ↓ | | | |
| **_Additional Staffing and Capacity_** | | | | | | | | | | | | | | | |
| Ratio of mental health provider per 5,000 patients | | | | | ↑ | | | | | | | | | | |
| Ratio of dental provider per 2,500 patients | | ↑ | ↑ | | ↑ | | | | | | | | ↑ | | |
| Ratio of enabling service staff per 5,000 patients | | | | | | | | | ↓ | | | | | | |
| Average number of services provided in addition to medical care | | ↓ | | | ↓ | | | | ↑ | | | | | | |
| **_Financial Resources_** | | | | | | | | | | | | | | | |
| Per capita total revenues | | | ↑ | | ↓ | | ↓ | | ↓ | | | ↑ | | ↑ | |
| Proportion of total revenues that are from 330 grants | | ↓ | ↓ | | | | | | | | | | ↓ | ↓ | |
| **_Contextual Characteristics_** | | | | | | | | | | | | | | | |
| Ratio of PCP per 5,000 population in county | ↑ | ↑ | ↑ | | | | | ↑ | | ↑ | | | | | |

(Continued)

**Table 3.** (Continued)

| | Prevention | | | | | | | | Care Management | | | | Outcomes | | |
|---|---|---|---|---|---|---|---|---|---|---|---|---|---|---|---|
| | Up-to-Date Childhood Immunization Completion | Receipt of Recommended Cervical Cancer Screening | Receipt of Colorectal Cancer Screening | Tobacco Use and Cessation Counseling and Intervention | Depression Screening and Receipt of a Follow-up Plan | Weight Assessment and Counseling for Nutrition and Physical Activity for Children and Adolescents | Body Mass Index (BMI) Screening and Follow-up Plan for adults | Patients with Asthma Receiving Appropriate Medications | Patients with Coronary Artery Diseases That Were Prescribed Lipid-Lowering Therapy | Patients with Ischemic Vascular Disease Who Used Aspirin or Another Antithrombotic Drug | Patient Seen for Follow-up Care within 90 Days of initial HIV Diagnosis | Pregnant Women Who Received Early Prenatal Care | Patients with Diabetes with Hemoglobin A1c Greater Than 9% | Patients with Hypertension with Blood Pressure below 140/90 | Patients Born Whose Birthweight Was Below Normal |
| Proportion below federal poverty guideline in county | | | | | | | | | | | | ↑ | | | |
| Proportion of minority in county | | | | | | | | | | | | | ↓ | | ↓ |

Notes:

↑ denotes positive statistically significant association at p<0.05.

↓ denotes negative statistically significant association at p<0.05.

Analyses were conducted using fractional outcome regression models using the logit distribution.

BMI, body mass index; CAD, coronary artery disease; IVD, ischemic vascular disease; HIV, human immunodeficiency virus; HbA1c, Hemoglobin A1c.

hypertension control may be because HCs with a higher concentration of these patients spent more intensive effort on improving these outcomes, or more diabetes patients sought care from these organizations if they offered diabetes specific services such as lifestyle or exercise classes [40, 41].

Among clinical quality measures, few control variables systematically explained urban/rural differences. Among outcome measures, the negative relationship of higher rates of racial/ethnic minority patients at the HC with poorer outcomes may have been because the racial/ethnic case mix captured social determinants of health that were not separately controlled for in our models. A number of studies have found similar results among the patient minority case mix and its effect on clinical performance, particularly on outcome measures [42, 43].

The only difference that remained significantly lower among rural HCs after adjusting for patient and HC characteristics was follow-up care among newly diagnosed HIV patients. This lower rate has been observed among low-income rural populations nationally and highlights a more pervasive challenge in rural areas [44, 45]. Other data indicate that these lower rates may be due to inadequately trained providers in rural areas to treat persons with HIV and distance or lack of readily available transportation to obtain services in rural areas [45, 46]. With the exception of the rural disparity in HIV performance measure, our data indicated that after controlling for patient and HC factors, there were no statistically significant differences in rural and urban HCs in their performance of clinical quality measures, including similar proportions that met or exceeded national HEDIS benchmarks. Both urban and rural HCs reported high achievement rates in meeting or exceeding national performance benchmarks in tobacco screening and cessation counseling, asthma treatment, lipid lowering therapy, poorly controlled diabetes, and hypertension control, and these performance rates, particularly outcome measures, are consistent with previous findings [42, 47, 48]. However, the performance achievement rates of both rural and urban HCs were low in other national benchmarks. Thus, patients of rural HCs may still experience disparities in quality of care [3, 42].

Our study had limitations including a single urban or rural designation for HC organizations even if some delivery sites may not be in rural areas. However, we used delivery site addresses to determine that 11% of sites among HCs that self-designated as rural may be urban and 13% of sites among HCs that self-designated as urban may be rural. This potential discrepancy is likely to be a consequence of variations in definitions of rural designation and the lack of UDS data on individual HC sites, which requires HCs to make an overall determination even if there is an urban and rural mix among the organization's sites. Given that our assessment found that potential misclassification is fairly uniform (11 and 13%), the bias that results is likely to weaken the associations between our outcomes of interest and urban and rural status. Additionally, because UDS data lacks information on individual HC sites, there is a potential masking of differences at site-level or patient-level. Our study is cross-sectional in nature and causal relationships between our independent and dependent variables cannot be readily determined. In addition, we examined the missing rate for the patients seen for follow-up care within 90 days of initial HIV diagnosis by urban/rural status and found that there is a positive association between this measure and rurality. This might be because HCs in rural areas have a low prevalence of patients with HIV and that these numbers were too small or sensitive to report, leading to a potential overestimate of the variations between urban and rural HCs for this outcome. Furthermore, it is possible that performance among clinical measures are independently correlated and are overestimated. Our national benchmarks are based on performance measures for Medicaid managed care organizations, which are a subset of HC patients and may limit national generalizability. However, because the majority of HC patients are Medicaid beneficiaries, these national benchmarks are likely to be the most relevant. Despite this limitation, both rural and urban HCs performed well in several preventive and outcome measures. Future

research should include several years of data to assess the role HC characteristics have in eliminating differences among urban and rural clinical performance over time.

## Policy implications

The number of rural populations has decreased over time and their demographics have shifted [1]. Our findings highlighted comparable clinical performance between urban and rural HCs, even with the cited challenges of providing care in rural geographies. These findings stress the integral role of rural HCs in alleviating disparities in quality of care and the potential negative impact of any reductions in resources to these crucial safety net providers in rural areas. Urban/rural disparities in HIV screening and follow-up requires further attention by assessing availability of trained providers in rural areas to treat persons with HIV, identifying procedures that improve confidentiality, or providing community health education to better inform the resident and provider communities about HIV, its epidemiology, and its implication for care and treatment [45, 49]. Improving availability of providers trained in HIV care in rural areas can be achieved by federal policies that are being implemented to improve access to care in rural areas with programs leveraging HCs to diagnose, treat, prevent, and respond to HIV in communities with substantial HIV burden [50]. HCs have received funding to provide medical care services to patients living with HIV through the Ryan White HIV/AIDS Program and in 2019, HRSA provided more than $2 billion to increase access to care for people living with HIV, including in rural areas [51]. Other programs including loan repayment programs under the National Health Service Corps for providers working in shortage areas, state-based loan repayment programs, and the Teaching Health Center Graduate Medical Education Program, which allow HCs to operate medical residency training programs, help address emerging public health priorities [17].

Other research indicates that lower clinical performance of HCs is linked to geographic disparities that could be alleviated by increasing the availability of resources and technical assistance [52]. HRSA has supported HC infrastructure development and provided funding to bolster the ability of these organizations to improve quality of care for low-income and uninsured patients [16, 53]. The Federal Office of Rural Health Policy has implemented programs to address access to quality health care and health professional capacity impacting rural communities. In addition, HRSA support of Health Center Controlled Networks and Primary Care Associations also provide technical resources to improve quality of care in rural and urban HCs [52, 54].

Our findings provide support for the continuation of these programs and the identification and implementation of new programs that address performance gaps among rural HCs. Promoting quality of care among rural and urban HCs could be achieved by providing technical assistance to develop skills and resources to conduct quality improvement activities [55]. Ultimately, our findings suggest that many urban/rural disparities in quality of care are concentrated among non-HC providers and further research is need to identify reasons for such disparities.

## Supporting information

**S1 Checklist. STROBE statement—Checklist of items that should be included in reports of *cross-sectional studies*.**
(DOC)

**S1 Dataset.**
(XLSX)

**S1 Table. Performance measures and full clinical quality performance measure definition.** (DOCX)

**S2 Table. Unadjusted and adjusted predicted probabilities of health center quality indicators by urban and rural status.** (DOCX)

**S3 Table. Regression models of prevention quality performance indicators.** (DOCX)

**S4 Table. Regression models of care management quality indicators.** (DOCX)

**S5 Table. Regression models of outcome quality indicators.** (DOCX)

**S6 Table. Logistic regression models of health centers that met prevention quality performance indicator benchmarks.** (DOCX)

**S7 Table. Logistic regression models of health centers that met care management quality performance indicator benchmarks.** (DOCX)

**S8 Table. Logistic regression models of health centers that met outcome quality performance indicator benchmarks.** (DOCX)

## Acknowledgments

### Disclaimer

The views expressed in this publication are solely the opinions of the authors and do not necessarily reflect the official policies of the U.S. Department of Health and Human Services or the Health Resources and Services Administration, nor does mention of the department or agency names imply endorsement by the U.S. Government.

## Author Contributions

**Conceptualization:** Nadereh Pourat, Xiao Chen, Connie Lu, Weihao Zhou, Hank Hoang, Alek Sripipatana.

**Data curation:** Connie Lu, Weihao Zhou.

**Formal analysis:** Nadereh Pourat, Xiao Chen, Weihao Zhou.

**Funding acquisition:** Nadereh Pourat.

**Investigation:** Nadereh Pourat.

**Methodology:** Nadereh Pourat, Xiao Chen.

**Project administration:** Connie Lu.

**Resources:** Connie Lu, Weihao Zhou.

**Software:** Connie Lu, Weihao Zhou.

**Supervision:** Nadereh Pourat.

**Writing – original draft:** Nadereh Pourat, Connie Lu.

**Writing – review & editing:** Xiao Chen, Weihao Zhou, Hank Hoang, Alek Sripipatana.

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
