## [Decision Letter · Decision Letter 0]

24 Jan 2020

PONE-D-19-27498

Assessing clinical quality performance and staffing capacity differences between urban and rural Health Resources and Services Administration-funded health centers in the United States: A cross sectional study

PLOS ONE

Dear Dr. Pourat,

Thank you for submitting your manuscript to PLOS ONE. After careful consideration, we feel that it has merit but does not fully meet PLOS ONE’s publication criteria as it currently stands. Therefore, we invite you to submit a revised version of the manuscript that addresses the points raised during the review process.

While both authors had positive comments about the topic and the appropriateness of the data used for the analysis, one reviewer had concerns about missing data that was present in the data file that was included with the submission. Please carefully review Reviewer #1's concerns about data management and respond. The same reviewer also had major concerns about the appropriateness of the statistical tests that were conducted, please review, re-do analyses as necessary and respond.

We would appreciate receiving your revised manuscript by Feb 24 2020 11:59PM. To enhance the reproducibility of your results, we recommend that if applicable you deposit your laboratory protocols in protocols.io, where a protocol can be assigned its own identifier (DOI) such that it can be cited independently in the future. For instructions see: http://journals.plos.org/plosone/s/submission-guidelines#loc-laboratory-protocols

We look forward to receiving your revised manuscript.

Kind regards,

PLOS ONE Academic Editor

Reviewers' comments:

Reviewer's Responses to Questions

**Comments to the Author**

1. Is the manuscript technically sound, and do the data support the conclusions?

Reviewer #1: Partly

Reviewer #2: Yes

2. Has the statistical analysis been performed appropriately and rigorously? 

Reviewer #1: No

Reviewer #2: Yes

3. Have the authors made all data underlying the findings in their manuscript fully available?

Reviewer #1: Yes

Reviewer #2: Yes

4. Is the manuscript presented in an intelligible fashion and written in standard English?

Reviewer #1: Yes

Reviewer #2: Yes

5. Review Comments to the Author

Reviewer #1: Summary: The authors first merged the data collected from 1,373 HRSA-funded Health Centers (HC) and county level data of the 2016 AHRF, and then proceeded to tabulate the various health quality indicators broken down by HC-reported urban/rural status. To discern the adjusted differences, outcome-specific regression models were built (adjusted for patient-level, organizational level, and county-level variables.) The authors found that among adjusted model, HIV linkage persisted to be lower in the rural HCs, while other urban/rural differences were explained away by the various independent variables, with proportions of non-English speakers, people living with diabetes, and people aged 17 or below, as well as PCP/population being the frequent candidates. The authors provided policy-level suggestions on such findings.

Major comments:

Overall, I really enjoyed reading the paper. The data choice was proper and the idea that urban/rural engulfs many different factors was well displayed. The policy-level recommendations were also moderately realistic that they could be referenced by interested policy stakeholders. However, I have some major concerns on result presentation and statistical analyses. I am listing them here for the authors’ reference and I hope addressing them may further strengthen their work.

The results in the supplementary tables should be in the text: First, nearly every statistically significant difference in Table 1 was verbatim mentioned in text, rendering Table 1 unnecessary; Second, Tables 2 and Table 3 are better combined so that the unadjusted and adjusted differences can be more readily compared and contrasted; Third, urban/rural differences in healthcare quality is well-reported but the factors contributing to that in a systematic analysis are not as well-known. If the authors can creatively visualize the significance levels and directions of the regression coefficients without resorting to showing large amount of numbers, then readers will be able to appreciate important interpretations like the one spanning from line 236 through 241.

The use of GLM deserves some clarification: “Generalized Linear Model” includes a lot of different statistical models, so please specify that in the Abstract. From the Methods (Line 189 through 196) it seems the authors set it as a logistic regression, if so, please just state that. Assuming the analysis was indeed logistics regression, why is it the best choice given the dependent variables were continuous percentages? Since there are also beta regression and probit regression, etc. which could be more suitable for these kinds of outcomes, it would be helpful if the authors could provide a method-related paper to justify the use of logistic regression in this fashion. Along the same line, if dichotomization was performed, then please provide the scheme.

How did the model address the difference sizes of HC? The overall analysis seems to assume equal weight for every HC, are they comparable in size (e.g. in terms of patients served)? If not, should the summary statistics and regression models be weighted?

Concerns on data management: The attached Excel data set shows that HIV linkage was missing in almost half of the HCs. Given HIV linkage was the only characteristic still found to be different in urban/rural settings, the prevalence of missing outcomes as well as some speculation are merited. In addition, some of the percentage data were shown in percentage, and yet some were shown in fraction (e.g. percent of patient with HIV.) This may explain why the regression coefficients for those covariates in fraction were much larger. I would suggest a round of audit to verify the data, software syntax, and output results.

Minor comments/suggestions:

[Abstract] The starting sentence created a false impression that HCs are exclusive to rural area, creating some confusion later. Please revise.

[Line 115 through 117] The study design, data, and analysis do not support this objective. First, it is cross-sectional so “contribute in reducing disparities” could be over-reaching; second, the analysis adjusted for many causal downstream variables of urban/rural, while I would agree that the work unpacked what urban/rural entails, the fact that the urban/rural became largely statistically non-significant is not indicative of reducing disparity, but perhaps mediation adjustment.

[Line 137 through 139] Are there only 15 in HEDIS? If there were more than 15, how did the authors decide on the final list?

[Line 194] The brand name should be written as Stata.

[Line 202] Add SD to the acronym list.

[Table 1 and others, including Excel file] Please check the label “Percent of patients of patients 65 years and older.”

[Table 3] HEDIS benchmarks should be accompanied by “percent of HCs that exceed the benchmark” rather than using only the sample mean to determine adherence.

Reviewer #2: This article uses the Uniform Data Set to compare HRSA funded clinics operating in urban and rural areas using a number of standard quality metrics. The authors find that after controlling for confounders, there are no statistically significant differences between urban and rural clinics for most outcome measures. The one exception is that Rural clinics "had lower rates of linking patients newly diagnosed with HIV to care."

Overall, this is a nice article that makes a useful contribution to the literature. As the authors point out, there are relatively few comparisons of this sort, which is surprising given the well know differences in urban-rural health outcomes. Despite this, I think the paper could benefit from a bit more reflection on both the premise of the study and the implications of these findings. Most studies documenting urban-rural differences in health status point to social determinants as an explanation. To the degree that there is a focus on health care, the usual emphasis is on the availability of care, not quality. Did the authors have reason to believe there might be differences in the quality (as opposed to the quantity) of care available to people living in urban and rural areas? This is an underlying assumption of the analysis that the authors do not set up well.

Second, I think the quality of care available in HRSA funded clinics is important, but given the relatively small role that such clinics play in the overall health system, would it be reasonable to suggest that differences in quality would be sufficient to explain urban-rural differences in the first place? Or are the authors focused more narrowly on urban-rural differences in health care for populations who are likely to seek care in HRSA clinics and other safety-net organizations?

Third, what should policy makers do with this information? If there is a concern about quality differences among HRSA clinics in urban and rural areas, the authors have offered comforting evidence -- but the urban rural health differences remain. So if it is not a quality difference, what's driving the problem?

Finally, I think the authors should say a bit more about their HIV finding. There was an article published in the NYT recently arguing that HIV is increasing in rural areas, but that these parts of the country are not ready for it. The findings in this paper are one small part of that, but the findings are certainly consistent with the concerns expressed in the article. I think the authors should put the HIV finding into the larger context of rural public health and health care capacity to address a growing HIV problem in these communities.

6. PLOS authors have the option to publish the peer review history of their article (what does this mean?). If published, this will include your full peer review and any attached files.

Reviewer #1: No

Reviewer #2: No

---

## [Author Response · Author response to Decision Letter 0]

4 Mar 2020

Reviewer Comment

Reviewer #1: 

Summary: The authors first merged the data collected from 1,373 HRSA-funded Health Centers (HC) and county level data of the 2016 AHRF, and then proceeded to tabulate the various health quality indicators broken down by HC-reported urban/rural status. To discern the adjusted differences, outcome-specific regression models were built (adjusted for patient-level, organizational level, and county-level variables.) The authors found that among adjusted model, HIV linkage persisted to be lower in the rural HCs, while other urban/rural differences were explained away by the various independent variables, with proportions of non-English speakers, people living with diabetes, and people aged 17 or below, as well as PCP/population being the frequent candidates. The authors provided policy-level suggestions on such findings.

Major comments:

Overall, I really enjoyed reading the paper. The data choice was proper and the idea that urban/rural engulfs many different factors was well displayed. The policy-level recommendations were also moderately realistic that they could be referenced by interested policy stakeholders. However, I have some major concerns on result presentation and statistical analyses. I am listing them here for the authors’ reference and I hope addressing them may further strengthen their work.

Reviewer Comment

The results in the supplementary tables should be in the text: First, nearly every statistically significant difference in Table 1 was verbatim mentioned in text, rendering Table 1 unnecessary; Second, Tables 2 and Table 3 are better combined so that the unadjusted and adjusted differences can be more readily compared and contrasted; Third, urban/rural differences in healthcare quality is well-reported but the factors contributing to that in a systematic analysis are not as well-known. If the authors can creatively visualize the significance levels and directions of the regression coefficients without resorting to showing large amount of numbers, then readers will be able to appreciate important interpretations like the one spanning from line 236 through 241.

Response to Reviewers

We have edited description of Table 1 in the Results to reduce length and only emphasize important findings. 

We have combined Tables 2 and 3 and reformatted to visualize the significant findings as recommended. 

We did not include the coefficients from the supplemental tables in the text. Instead we reported the overall predicted probabilities that show the degree to which a given outcome differs between urban and rural HCs. We then supplemented that discussion with what indicators in models contributed to the outcomes. We believe that it was a more efficient way of describing the results of the extensive models and less confusing to the readers. We added a table that highlighted the significant coefficients for each quality indicator and whether they were positive or negative in response to this comment.

Location in Manuscript

Results, Page 12-13, line 254-263

Revised Table 2 (previously Tables 2 and 3)

Table 3 (new)

Reviewer Comment

The use of GLM deserves some clarification: “Generalized Linear Model” includes a lot of different statistical models, so please specify that in the Abstract. From the Methods (Line 189 through 196) it seems the authors set it as a logistic regression, if so, please just state that. Assuming the analysis was indeed logistics regression, why is it the best choice given the dependent variables were continuous percentages? Since there are also beta regression and probit regression, etc. which could be more suitable for these kinds of outcomes, it would be helpful if the authors could provide a method-related paper to justify the use of logistic regression in this fashion. Along the same line, if dichotomization was performed, then please provide the scheme.

Response to Reviewers

The underlying data of the dependent variables are presented as a proportion. For ease of interpretation, the dependent variables are transformed to proportions ranging between zero and one. We used a generalized linear model, not a logistic regression, and have made edits to the Methods to make this clear. We have revised the Methods to make clear the dependent variables are proportions and added detail to the Statistical Analysis section on how we performed this transformation.

Location in Manuscript

Abstract, Page 4, line 69

Methods, Dependent variables, page 9, line 167

Methods, Statistical analysis, Page 11, line 231

Reviewer Comment

How did the model address the difference sizes of HC? The overall analysis seems to assume equal weight for every HC, are they comparable in size (e.g. in terms of patients served)? If not, should the summary statistics and regression models be weighted?

Response to Reviewers 

Health centers are not comparable in size and as a result, we have controlled for the average number of sites and patients seen during the year in each model. We have also controlled for a number of patient characteristics. The full regression models with control variables are presented in the Supplemental Appendices. We are using the entire population of health centers in 2017 (N=1,373), not a sample.

Location in Manuscript

S2-S4 Tables

Reviewer Comment

Concerns on data management: The attached Excel data set shows that HIV linkage was missing in almost half of the HCs. Given HIV linkage was the only characteristic still found to be different in urban/rural settings, the prevalence of missing outcomes as well as some speculation are merited. In addition, some of the percentage data were shown in percentage, and yet some were shown in fraction (e.g. percent of patient with HIV.) This may explain why the regression coefficients for those covariates in fraction were much larger. I would suggest a round of audit to verify the data, software syntax, and output results.

Response to Reviewers

All clinical quality measure data are displayed as proportions, including HIV linkage to care. We have assessed the missing HIV linkage to care outcomes and found there was a relationship with urban/rural settings. We have added a sentence in the Limitations to address this. We further explain that missingness among rural health centers may be due to low prevalence of HIV and could be too sensitive to report and suggest a conservative interpretation. 

We have also confirmed in the software syntax the percentage data displayed in the outcome variables were rescaled as a fraction.

Location in Manuscript

Discussion, page 31, lines 429-435

Methods, Page 9-10, lines 166-167

Reviewer Comment

Minor comments/suggestions:

[Abstract] The starting sentence created a false impression that HCs are exclusive to rural area, creating some confusion later. Please revise.

Response to Reviewers

We have revised this sentence to indicate that health centers are located nationally.

Location in Manuscript

Abstract, Page 4, line 59-62

Reviewer Comment

[Line 115 through 117] The study design, data, and analysis do not support this objective. First, it is cross-sectional so “contribute in reducing disparities” could be over-reaching; second, the analysis adjusted for many causal downstream variables of urban/rural, while I would agree that the work unpacked what urban/rural entails, the fact that the urban/rural became largely statistically non-significant is not indicative of reducing disparity, but perhaps mediation adjustment.

Response to Reviewers

We have revised this sentence to be more conservative and consistent with our study design and analysis.

Location in Manuscript

Introduction, Page 6, line 95-100

Reviewer Comment

[Line 137 through 139] Are there only 15 in HEDIS? If there were more than 15, how did the authors decide on the final list?

Response to Reviewers

There are a total of 15 clinical quality measures that health centers are required to report and we decided to use all 15 measures. These 15 measures have benchmarks in HEDIS. We have revised for clarity. 

Location in Manuscript

Methods, Page 8, line 158-163

Reviewer Comment

[Line 194] The brand name should be written as Stata.

Response to Reviewers

Revised as suggested.

Location in Manuscript

Methods, Page 12, line 240

Reviewer Comment 

[Line 202] Add SD to the acronym list.

Response to Reviewers

We have added to the acronym list as suggested.

Location in Manuscript

Title Page, Page 3, line 57

Reviewer Comment 

[Table 1 and others, including Excel file] Please check the label “Percent of patients of patients 65 years and older.”

Response to Reviewers

Revised for correctness.

Location in Manuscript

Table 1 and S2 Table

Reviewer Comment 

[Table 3] HEDIS benchmarks should be accompanied by “percent of HCs that exceed the benchmark” rather than using only the sample mean to determine adherence.

Response to Reviewers

We have revised Table 2 to include the proportion of health centers that achieved HEDIS benchmarks.

Location in Manuscript

Revised Table 2 (previously Table 2 and 3)

Methods, Statistical Analysis

Reviewer Comment 

Reviewer #2: This article uses the Uniform Data Set to compare HRSA funded clinics operating in urban and rural areas using a number of standard quality metrics. The authors find that after controlling for confounders, there are no statistically significant differences between urban and rural clinics for most outcome measures. The one exception is that Rural clinics "had lower rates of linking patients newly diagnosed with HIV to care."

Overall, this is a nice article that makes a useful contribution to the literature. As the authors point out, there are relatively few comparisons of this sort, which is surprising given the well know differences in urban-rural health outcomes. Despite this, I think the paper could benefit from a bit more reflection on both the premise of the study and the implications of these findings. Most studies documenting urban-rural differences in health status point to social determinants as an explanation. To the degree that there is a focus on health care, the usual emphasis is on the availability of care, not quality. Did the authors have reason to believe there might be differences in the quality (as opposed to the quantity) of care available to people living in urban and rural areas? This is an underlying assumption of the analysis that the authors do not set up well.

Response to Reviewers

There is literature that shows differences in quality of care outcomes in the general urban/rural population, as well as other providers. Due to this evidence, we believe in examining urban/rural differences in health center settings in particular. We have added additional sentences in the Introduction in order to address differences in quality

Location in Manuscript

Introduction, Page 6, line 106-108

Reviewer Comment 

Second, I think the quality of care available in HRSA funded clinics is important, but given the relatively small role that such clinics play in the overall health system, would it be reasonable to suggest that differences in quality would be sufficient to explain urban-rural differences in the first place? Or are the authors focused more narrowly on urban-rural differences in health care for populations who are likely to seek care in HRSA clinics and other safety-net organizations?

Response to Reviewers

HCs are the cornerstone of the safety net system and the only providers that provide care to low-income and uninsured patients regardless of income or any other factor. Other providers including public systems organized by counties have various restrictions. In addition, HCs are often the only comprehensive provider of care in rural areas. Therefore, the role of HCs in providing access to high quality care in general and in rural areas in particular is very important.

 We are focused on urban/rural differences among populations that are served by HRSA-funded health centers, but our results have implications for quality of care in urban and rural areas more broadly, particularly when HCs are the primary or only providers of care

Location in Manuscript

Introduction, Page 7, line 123-125

Reviewer Comment 

Third, what should policy makers do with this information? If there is a concern about quality differences among HRSA clinics in urban and rural areas, the authors have offered comforting evidence -- but the urban rural health differences remain. So if it is not a quality difference, what's driving the problem?

Response to Reviewers

We cannot explain what explain urban/rural differences in quality from our data. However, we have shown that a subset of rural populations have quality of care on par with urban areas. Therefore, policy efforts should be focused on other providers who are likely to be the source of urban/rural disparities. We have revised the implications to address this point.

Location in Manuscript

Discussion, Page 34, line 479-481

Reviewer Comment 

Finally, I think the authors should say a bit more about their HIV finding. There was an article published in the NYT recently arguing that HIV is increasing in rural areas, but that these parts of the country are not ready for it. The findings in this paper are one small part of that, but the findings are certainly consistent with the concerns expressed in the article. I think the authors should put the HIV finding into the larger context of rural public health and health care capacity to address a growing HIV problem in these communities.

Response to Reviewers

We have added additional information to support how health centers have provided additional funding support to address HIV.

Location in Manuscript

Discussion, Page 33, line 458-460

---

## [Decision Letter · Decision Letter 1]

31 Jul 2020

PONE-D-19-27498R1

Assessing clinical quality performance and staffing capacity differences between urban and rural Health Resources and Services Administration-funded health centers in the United States: A cross sectional study

PLOS ONE

Dear Dr. Pourat,

Thank you for submitting your manuscript to PLOS ONE. After careful consideration, we feel that it has merit but does not fully meet PLOS ONE’s publication criteria as it currently stands. Therefore, we invite you to submit a revised version of the manuscript that addresses the points raised during the review process.

Editor's Comments: 

Given that the concerns of Reviewer 1 remain significant, and Reviewer 2 was unable to provide a second review, we have worked with the Editorial office to find another reviewer with a strong quantitative background. In your revision, please be sure to address the areas of overlap of both reviewers (1 & 3), as they share similar concerns about the statistical methodology.

Thank you.

We look forward to receiving your revised manuscript.

Kind regards,

Academic Editor

PLOS ONE

Reviewers' comments:

Reviewer's Responses to Questions

**Comments to the Author**

1. If the authors have adequately addressed your comments raised in a previous round of review and you feel that this manuscript is now acceptable for publication, you may indicate that here to bypass the “Comments to the Author” section, enter your conflict of interest statement in the “Confidential to Editor” section, and submit your "Accept" recommendation.

Reviewer #1: (No Response)

Reviewer #3: (No Response)

2. Is the manuscript technically sound, and do the data support the conclusions?

Reviewer #1: Partly

Reviewer #3: Yes

3. Has the statistical analysis been performed appropriately and rigorously? 

Reviewer #1: No

Reviewer #3: Yes

4. Have the authors made all data underlying the findings in their manuscript fully available?

Reviewer #1: Yes

Reviewer #3: Yes

5. Is the manuscript presented in an intelligible fashion and written in standard English?

Reviewer #1: Yes

Reviewer #3: Yes

6. Review Comments to the Author

Reviewer #1: Major comments: Thank you for addressing most of my comments concerning the presentation of the results. Tables 2 & 3 appear informative and I hope readers will find the results more readily digestible.

There are a few remaining issues that the authors did briefly responded to and I would like to raise them again:

It appears that the GLM analysis was based on binomial family and logit link, which makes this a logistic regression. After reading the authors’ response, my curiosity remains unsated and I wish the authors will address this explicitly: why was logistic regression used on a continuous variable that is bound between 0 and 1? It would be great if the author can cite a technical document that this is indeed a correct approach. The fractional dependent variables (like pct_diabete) look very much like a bell curve and I couldn’t wrap my head around using binomial. If we tried to use the Stata command “logit” or “logistic” to repeat this analysis we may also find that Stata would in fact display a warning and refuse to proceed. Please also consider if fracreg (Fractional response regression) function in Stata would be a more appropriate candidate.

With the Excel data, I was able to check some statistics in Table 1. It seems the data remained unweighted. Given the mean (SD) patient of 19,792 (23,663), there is a huge variability among the sizes. If a HC serving 1,000 patients has and indicator at 0%, and a HC serving 20,000 patients has it at 50%, would the mean be closer to 25.0% or 47.6%? It’d be great if the authors can justify why they favored the unweighted approach.

Reviewer #3: The analyses in this manuscript appear to be done well. I had some suggestions below to strengthen the descriptions and, potentially, improve the results. I think the biggest problem is the manuscript is so dense. Table 2 and the results section have quite a bit of information. I have a couple suggestions for this in my comments which may or may not help.

1. (line 172) This statement is a little confusing since an indicator variable is 0 or 1 and not a proportion. I'm guessing created indicator variables for each of the 15 DVs and then created a proportion from those? That would fit with your methods.

2. (lines 205-206) I think this choice of method is good. I suggest including a methodological citation for the method, probably McCullagh and Nelder's book will be fine. I think the main question is what your residuals look like and whether there is potential for systematic over- or under-fitting. My guess is that your covariates have eliminated this, but it would be good for you to check this. It may be that adding random effects by state to account for any state-to-state differences could be helpful. Another thought were spatial random effects, but my guess is that the urban/rural variable will be good enough for that.

3. (line 210) How many or what percentage of HCs were dropped due to missing data? Rules of thumb vary on when complete case analyses are still valid. My suggestion is if > 5% are dropped, then start exploring the missing data. If > 15% are missing, then you'll probably need to do something about it, e.g., multiple imputation.

4. (line 211) Sorry, I'm not a Stata user. Does the margins command produce confidence intervals or prediction intervals?

5. Did you perform any variable selection or were all the IVs included in all analyses?

6. (Table 2) This table is quite dense and I wonder if you thought about moving some of these results to a figure, especially the predicted probabilities. It might make for a nice visual and could make table 2 more readable. I also wondered if there was any way to abbreviate the measures in the first column.

7. (Table S2) I am not sure I understand the coefficients in this table. Are they the raw coefficient values from the binomial models? If so, they should be exponentiated to become odds ratios and reported as such.

7. PLOS authors have the option to publish the peer review history of their article (what does this mean?). If published, this will include your full peer review and any attached files.

Reviewer #1: No

Reviewer #3: No

---

## [Author Response · Author response to Decision Letter 1]

29 Sep 2020

Reviewer #1

Reviewer Comment

Major comments: Thank you for addressing most of my comments concerning the presentation of the results. Tables 2 & 3 appear informative and I hope readers will find the results more readily digestible.

There are a few remaining issues that the authors did briefly responded to and I would like to raise them again:

It appears that the GLM analysis was based on binomial family and logit link, which makes this a logistic regression. After reading the authors’ response, my curiosity remains unsated and I wish the authors will address this explicitly: why was logistic regression used on a continuous variable that is bound between 0 and 1? It would be great if the author can cite a technical document that this is indeed a correct approach. The fractional dependent variables (like pct_diabete) look very much like a bell curve and I couldn’t wrap my head around using binomial. If we tried to use the Stata command “logit” or “logistic” to repeat this analysis we may also find that Stata would in fact display a warning and refuse to proceed. Please also consider if fracreg (Fractional response regression) function in Stata would be a more appropriate candidate.

Response to Reviewers 

We used GLM regressions for the analyses because the dependent variables are a proportion bound between 0 and 1 and therefore had both a lower and upper bound ceiling. The following citation supports our rationale for using the GLM analysis (Baum CF. Stata Tip 63: Modeling Proportions. The Stata Journal. 2008;8(2):299-303. doi:10.1177/1536867X0800800212). In response to the reviewer’s comment we changed the regressions to fracreg even though our tests indicated the two models yielded the exact same results. We included a technical citation to support use of fracreg in the manuscript. 

Location in Manuscript

Methods, Page 11, line 209-210

Reviewer Comment

With the Excel data, I was able to check some statistics in Table 1. It seems the data remained unweighted. Given the mean (SD) patient of 19,792 (23,663), there is a huge variability among the sizes. If a HC serving 1,000 patients has and indicator at 0%, and a HC serving 20,000 patients has it at 50%, would the mean be closer to 25.0% or 47.6%? It’d be great if the authors can justify why they favored the unweighted approach.

Response to Reviewers

We did not weight the data because the goal of this manuscript was to compare the performance of HC in urban and rural locations. For this goal, every HC would have to exert the same amount of influence on the results. As the reviewer points out, the weighted data would allow larger HCs to exert more of an influence on the results. That analyses would be more appropriate if our aim was to compare the population-level impact. 

To illustrate the difference between weighted and unweighted data, we conducted weighted regressions to assess how the results change. We found the majority of coefficients had similar effects in both types of analyses, with the exception of HIV linkage to care and low birthweight. In weighted analyses, HIV linkage to care was no longer significant, but the direction of association did not change. In the weighted model, low birthweight became significant but the direction of the associations did not change. 

Location in Manuscript

No change.

Reviewer #3

Reviewer Comment

The analyses in this manuscript appear to be done well. I had some suggestions below to strengthen the descriptions and, potentially, improve the results. I think the biggest problem is the manuscript is so dense. Table 2 and the results section have quite a bit of information. I have a couple suggestions for this in my comments which may or may not help.

Response to Reviewers

We thank the reviewer for their comments. We have added Figure 1 to visually display the significant differences in performance measures between urban and rural HCs. We did not include results that were not significant to reduce the size of this figure. We moved the original Table 2 into the appendix to all performance measures . 

Location in Manuscript

Figure 1 (new)

Table 2 

Reviewer Comment

1. (line 172) This statement is a little confusing since an indicator variable is 0 or 1 and not a proportion. I'm guessing created indicator variables for each of the 15 DVs and then created a proportion from those? That would fit with your methods.

Response to Reviewers

We have removed this sentence to reduce confusion. The dependent variables are indeed proportions bound between 0 and 1 and therefore had both a lower and upper bound ceiling. They indicate the proportion of patients that had received specific services or had a specific outcome. The methods sections are revised in response to both reviewers’ comments.

Location in Manuscript

Methods, Page 8, line 158

Reviewer Comment

2. (lines 205-206) I think this choice of method is good. I suggest including a methodological citation for the method, probably McCullagh and Nelder's book will be fine. I think the main question is what your residuals look like and whether there is potential for systematic over- or under-fitting. My guess is that your covariates have eliminated this, but it would be good for you to check this. It may be that adding random effects by state to account for any state-to-state differences could be helpful. Another thought were spatial random effects, but my guess is that the urban/rural variable will be good enough for that.

Response to Reviewers

As indicated in the previous responses to similar comments above, we are now using fracreg or fractional outcome regression, which produced the same results as GLM and have added a citation to support the use of this regression model. 

We have also checked the residual plots for all our models and found that there was no evidence of model under- or overfitting. 

We have controlled for county-level variations using county-level characteristics, as well as the urban/rural variable as the reviewer mentioned. We have not included random effects by state to avoid any potential overfitting/overcontrolling.

Location in Manuscript 

Methods, Page 11, lines 209-210

Reviewer Comment 

3. (line 210) How many or what percentage of HCs were dropped due to missing data? Rules of thumb vary on when complete case analyses are still valid. My suggestion is if > 5% are dropped, then start exploring the missing data. If > 15% are missing, then you'll probably need to do something about it, e.g., multiple imputation.

Response to Reviewers

We examined the proportion of missing for each dependent variable and found that no variables were missing by greater than 5%. We did not impute the dependent variables with missing values following recommendation from Hippel (Von Hippel, P.T. (2007), REGRESSION WITH MISSING YS: AN IMPROVED STRATEGY FOR ANALYZING MULTIPLY IMPUTED DATA. Sociological Methodology, 37: 83-117. doi:10.1111/j.1467-9531.2007.00180.x). 

Location in Manuscript 

No change

Reviewer Comment 

4. (line 211) Sorry, I'm not a Stata user. Does the margins command produce confidence intervals or prediction intervals?

Response to Reviewers

The margins command produces predicted probabilities with confidence intervals. 

Location in Manuscript 

S2 Table 

Reviewer Comment 

5. Did you perform any variable selection or were all the IVs included in all analyses?

Response to Reviewers

We had selected independent variables that could influence HC performance conceptually and included all of them in the models to avoid omitted variable bias. 

Location in Manuscript 

No change

Reviewer Comment 

6. (Table 2) This table is quite dense and I wonder if you thought about moving some of these results to a figure, especially the predicted probabilities. It might make for a nice visual and could make table 2 more readable. I also wondered if there was any way to abbreviate the measures in the first column.

Response to Reviewers

We agree that the table was dense. This was the result of requests from previous reviewers to present the data in that way. We have now presented some of the data in the original Table 2 into Figure 1 as suggested and abbreviated the name of the measures for better readability. We then adjusted the other tables including S2 Table and Table 2 still reports probabilities of met or exceeding benchmarks. 

Location in Manuscript 

Figure 1 (new)

S2 Table

Table 2 

Reviewer Comment 

7. (Table S2) I am not sure I understand the coefficients in this table. Are they the raw coefficient values from the binomial models? If so, they should be exponentiated to become odds ratios and reported as such.

Response to Reviewers

Yes, the supplemental data displayed beta coefficients. We have now revised them to report odds ratios as suggested. 

Location in Manuscript 

S3 Table- S5 Table

---

## [Decision Letter · Decision Letter 2]

2 Nov 2020

PONE-D-19-27498R2

Assessing clinical quality performance and staffing capacity differences between urban and rural Health Resources and Services Administration-funded health centers in the United States: A cross sectional study

PLOS ONE

Dear Dr. Pourat,

Thank you for submitting your manuscript to PLOS ONE. After careful consideration, we feel that it has merit but does not fully meet PLOS ONE’s publication criteria as it currently stands. Therefore, we invite you to submit a revised version of the manuscript that addresses the points raised during the review process.

Thank you for addressing the comments of our reviewers. While both reviewers indicated that they are ready to accept the manuscript, Reviewer One requested that an additional statement be added to the Methods section indicating that all HC's were treated with equal analytical weight. Please add this statement where appropriate and resubmit the manuscript with this revision.

We look forward to receiving your revised manuscript.

Kind regards,

Candace C. Nelson

Academic Editor

PLOS ONE

Reviewers' comments:

Reviewer's Responses to Questions

**Comments to the Author**

1. If the authors have adequately addressed your comments raised in a previous round of review and you feel that this manuscript is now acceptable for publication, you may indicate that here to bypass the “Comments to the Author” section, enter your conflict of interest statement in the “Confidential to Editor” section, and submit your "Accept" recommendation.

Reviewer #1: (No Response)

Reviewer #3: All comments have been addressed

2. Is the manuscript technically sound, and do the data support the conclusions?

Reviewer #1: Yes

Reviewer #3: (No Response)

3. Has the statistical analysis been performed appropriately and rigorously? 

Reviewer #1: Yes

Reviewer #3: (No Response)

4. Have the authors made all data underlying the findings in their manuscript fully available?

Reviewer #1: Yes

Reviewer #3: (No Response)

5. Is the manuscript presented in an intelligible fashion and written in standard English?

Reviewer #1: Yes

Reviewer #3: (No Response)

6. Review Comments to the Author

Reviewer #1: Thank you for addressing the previous comments by adding the technical reference and explaining the paradigm between weighted and unweighted analysis. I just have one minor suggested change: in the Methods section please indicate that all HCs were treated with equal analytical weight.

Reviewer #3: (No Response)

7. PLOS authors have the option to publish the peer review history of their article (what does this mean?). If published, this will include your full peer review and any attached files.

Reviewer #1: No

Reviewer #3: No

---

## [Author Response · Author response to Decision Letter 2]

9 Nov 2020

Reviewer #1

Reviewer Comment

Thank you for addressing the previous comments by adding the technical reference and explaining the paradigm between weighted and unweighted analysis. I just have one minor suggested change: in the Methods section please indicate that all HCs were treated with equal analytical weight.

Response to Reviewers

The authors have included a sentence in the Methods as suggested. 

Location in Manuscript

Methods, Line 211, page 11

---

## [Editor Report · Decision Letter 3]

11 Nov 2020

Assessing clinical quality performance and staffing capacity differences between urban and rural Health Resources and Services Administration-funded health centers in the United States: A cross sectional study

PONE-D-19-27498R3

Dear Dr. Pourat,

We’re pleased to inform you that your manuscript has been judged scientifically suitable for publication and will be formally accepted for publication once it meets all outstanding technical requirements.

Kind regards,

Candace C. Nelson, ScD

Academic Editor

PLOS ONE

---

## [Editor Report · Acceptance letter]

24 Nov 2020

PONE-D-19-27498R3 

Assessing clinical quality performance and staffing capacity differences between urban and rural Health Resources and Services Administration-funded health centers in the United States: A cross sectional study 

Dear Dr. Pourat:

I'm pleased to inform you that your manuscript has been deemed suitable for publication in PLOS ONE. Congratulations! Your manuscript is now with our production department. 

Kind regards, 

on behalf of

Dr. Candace C. Nelson 

Academic Editor

PLOS ONE